# Human Astrovirus Outbreak in a Daycare Center and Propagation among Household Contacts

**DOI:** 10.3390/v13061100

**Published:** 2021-06-09

**Authors:** Ignacio Parrón, Elsa Plasencia, Thais Cornejo-Sánchez, Mireia Jané, Cristina Pérez, Conchita Izquierdo, Susana Guix, Àngela Domínguez

**Affiliations:** 1Sub-Direcció Regional a Barcelona del Departament de Salut, 08005 Barcelona, Spain; elsa.plasencia@gencat.cat (E.P.); cperezca@gencat.cat (C.P.); 2Departament de Medicina, Universitat de Barcelona, 08036 Barcelona, Spain; mireia.jane@gencat.cat (M.J.); angela.dominguez@ub.edu (À.D.); 3Departament de Microbiologia, Vall d’Hebrón Hospital, 08035 Barcelona, Spain; thais.cornejo@vhir.org; 4Sub-Direcció General de Vigilància i Resposta a Emergències de Salut Pública, 08005 Barcelona, Spain; conchita.izquierdo@gencat.cat; 5CIBER Epidemiologia y Salud Pública, Instituto de Salud Carlos III, 28029 Madrid, Spain; 6Grup de Virus Entérics, Departament de Genètica Microbiologia i Estadística, Universitat de Barcelona, 08028 Barcelona, Spain; susanaguix@ub.edu; 7Institut de Recerca en Nutrició i Seguretat Alimentària (INSA-UB), Universitat de Barcelona, 08921 Santa Coloma de Gramenet, Spain

**Keywords:** astrovirus, outbreak, acute gastroenteritis, daycare center, household contacts

## Abstract

We investigated an outbreak of acute gastroenteritis due to human astrovirus in a daycare center, describing the transmission mechanism, the most affected age groups, conditioning factors and the extent of the outbreak among household contacts of the daycare center attenders. Data were collected from persons exposed at the daycare center and their home contacts. Fecal samples from affected and non-affected daycare center attenders were analyzed for viruses causing acute gastroenteritis by RT-PCR. The percentage of households affected and the attack rates (AR) were calculated. The attack rates were compared using the rate ratio (RR) with 95% confidence intervals. Information was obtained from 245 people (76 attenders and 169 contacts) of whom 49 were clinical cases. Five HAstV-4, two HAstV-8 and three non-typable HAstV cases were identified (six from clinical cases and four from asymptomatic infected people). The global AR was 20% (41.2% in children aged < 2 years). Data were obtained from 67 households: 20 households of affected attenders and 47 of non-affected attendees. Household contacts of affected attenders had a higher AR (74.3%) than that of non-affected attendees (2.4%). We found asymptomatic infections amongst daycare attendees. The transmission of HAstV during the outbreak was not limited to the daycare center but extended to household contacts of both affected and non-affected attenders.

## 1. Introduction

Human astroviruses (HAstV) are non-enveloped RNA viruses that cause up to 10% of sporadic cases of acute gastroenteritis (AGE) in children aged < 3 years [1,2,3] and are the third or fourth cause of viral AGE in children of this age [4,5,6,7].

Eight serotypes (HAstV-1 to HAstV-8) of HAstV are known to cause gastroenteritis. HAstV-1 is the most-frequently detected serotype worldwide [1,8,9], although other serotypes have occasionally been shown to be more prevalent in some geographical areas [4]. Infection with one serotype seems not to confer immunity against other serotypes [3].

Astroviruses are usually spread through direct person-to-person transmission via the fecal-oral route [10], although indirect transmission due to exposure to contaminated food or water has been described [11]. Astroviruses cause outbreaks, especially in healthcare [12,13] and daycare centers [14,15], mainly affecting small children.

Seroprevalence studies show that most children have been infected with HAstV by the age of 6 [16] and 75% of children aged 5–10 have antibodies [17]. More than 50% of children with diarrheal symptoms in whom HAstV are detected are aged < 2 years [6,14]. Asymptomatic infections are frequent [18], especially in daycare centers, and may represent more than 50% of persons infected [14]. To our knowledge, the involvement of home contacts in HAstV daycare center outbreaks has not been previously reported.

The objective of this study was to investigate an outbreak of AGE due to HAstV in a daycare center and determine the mode of transmission, the most-affected age groups and the spread to household contacts of attendees (children and workers) at the daycare center.

## 2. Materials and Methods

### 2.1. Study Design

On 9 March 2017, the Public Health Agency of Catalonia was notified of a possible AGE outbreak affecting three children who had initiated symptoms in the previous 48 h in a daycare center.

The center was attended by 110 children aged 0–3 years and 13 workers. Meals were served by an external catering company because the center had no kitchen of its own.

The epidemic period was defined as 1 February to 12 March 2017, and a clinical case was defined as any person in the daycare center or their household contacts who initiated symptoms of nausea, vomiting or diarrhea during the epidemic period. Confirmed cases were clinical cases in whom HAstV was detected by real time reverse transcription polymerase chain reaction (RT-PCR) in stool samples. A household contact was defined as an individual residing in the same home as a child or worker from the daycare center during the epidemic period.

An epidemiological survey was made that included demographic data, the date of onset and end of clinical manifestations and the symptoms presented, the number of people living in the home of each case, their age, whether they became ill and, if so, the date of symptom onset. Data were collected using a self-administered questionnaire that was completed by the parents or by the center workers.

A stool sample was collected from center attendees for microbiological study between March 13 and March 15 and sent to the Clinical Microbiology Laboratory of Vall d’Hebron Hospital. No samples were collected from household contacts.

According to clinical and epidemiological information, a viral etiology was suspected, and no bacteriological study was carried out. NucliSENS^®^ easyMAG^®^ (Biomerieux, Marcy-l’Étoile, France) was used to obtain genetic material. RT-PCR was used to identify the main viruses causing AGE (norovirus GI and GII, rotavirus, adenovirus, astrovirus and sapovirus) using the Allplex GI-Virus Assay (Seegene Inc. Seoul, Korea). Astrovirus genotyping was performed by sequencing the RT-PCR product according to a previously described protocol [19], and compared with Genbank sequences using Blastn analysis, which was also used to compare isolates within the same genotype.

### 2.2. Data Processing and Analysis

Data were collected using Microsoft Access (version 14.0) for cleansing and analysis with the PASW Statistic 18.2.0 software package and Epi Info™ for Windows.

The epidemic curve for cases at the daycare center and in home contacts was constructed to indicate the possible transmission mechanism.

The median age of affected and non-affected persons and the frequency of symptoms were calculated. The distribution of stool sample results by age groups was analyzed.

### 2.3. Statistical Analysis

The mean age of affected and non-affected persons in the daycare center and affected and non-affected household contacts was compared using the Student’s *t*-test. Statistical significance was established as *p* < 0.05.

Attack rates (AR) were calculated for daycare center attenders (globally and by age groups) and their household contacts (globally, by affected or non-affected attenders and by age group of attenders). The AR of the categories in each group were compared using the rate ratio (RR) and 95% confidence intervals (CI).

## 3. Results

A total of 76 daycare center attenders of (71 children and 5 workers) responded to the survey, representing 61.8% of those exposed, and information was obtained from 169 household contacts. In total, information was obtained from 245 subjects, 49 of whom met the clinical-case definition (18 children, 2 workers and 29 household contacts). No affected person required hospital admission.

The first person (case 1) affected in the daycare center was a care assistant whose date of symptom onset was February 12. Her 6-year old son had initiated symptoms two days before. The care assistant’s stool sample was negative.

Between the first and second case in the daycare center (case 4), two other people related with the center presented symptoms: the mother of a girl attending the daycare center (case 2) (from whom no stool sample was obtained) and the brother (case 3) of another girl from the same class. The second case at the center was a 2-year-old girl from the same class as the first case, who initiated symptoms on February 25; no stool sample was obtained (Figure 1). Nineteen stool samples were analyzed (13 from affected attenders and 6 from non-affected attenders).

The overall AR was 26.3% (20/76) for daycare center attenders and 17.2% (29/169) in household contacts. Considering daycare center attenders and their household contacts jointly, the AR was 20% (49/245), 17.2% in males (22/128) and 23.1% in females (27/117) (RR 0.74 95 CI 0.45, 1.23).

The mean age was 1 year 237.6 days (SD 176.5) in affected child attenders at the daycare center and 2 years and 54.6 days (SD 287.4) in non-affected child attenders (*p* = 0.002).

The distribution of AR and RR according to age are shown in Table 1. The most affected age group in attenders were children aged 1 year. The ARs were lower in other groups and the differences were only significant in children aged 2 years.

The mean age was 26 years (SD 17 years) in affected contacts and 29.4 years (SD 13.9 years) in non-affected contacts (*p* = 0.25).

In household contacts the AR by age groups did not differ significantly.

All persons in the daycare center who were affected presented diarrhea, 20% vomiting, 15% abdominal pain and 15% fever. Symptoms lasted 3.8 days on average (SD 2.4 days).

All samples were negative for rotavirus, adenovirus, sapovirus and norovirus GII. Ten samples (six from affected children, three from non-affected children and one from non-affected childcare staff) were positive for astroviruses (five HAstV-4, two HAstV-8 and three non-typeable) (Table 2). Sequence comparison with HAstV reference strains showed a 98.52% and 98.52% nucleotide identity with EF138831.1 and MG970100.1, respectively. All sequences belonging to the same serotype were identical. Sequences showed the closest similarity to published sequences.

Norovirus genogroup I was identified in two samples from daycare attenders (one from an affected person and one from a non-affected person). In household contacts of non-affected persons and in whom HAstV was detected in stools (two HAstV-4 and two HAstV-8) there were no symptoms of illness.

Sixty-seven of the 76 daycare attenders provided information about their household contacts: 44 household contacts of 20 affected attenders and 125 household contacts of 47 non-affected attenders.

There was a close association between clinical manifestations in attenders and household contacts (RR 30.95%; 95% CI 9.95–96.27) (Table 3) although clinical cases were also reported in household contacts of non-affected attenders (AR 2.4%)

There were no secondary clinical cases among the contacts of children aged < 1 year; the AR was 29.2% in contacts of 1-year-old children, 8.6% in contacts of 2-year-old children, 9.5% in contacts of 3-year-old children and 33.3% in contacts of workers (Table 4). Statistically significant differences were observed in the AR of household contacts of 1- and 2-year-old children (RR 3.04 95% CI 1.51–6.12 and RR 0.40 95% CI 0.16–0.99, respectively).

## 4. Discussion

According to published reports, this is the first descriptive study of an outbreak of AGE with circulation of several serotypes of HAstVs due to person-to-person transmission in a daycare center in which household contacts were also studied.

Astrovirus is responsible for 2–9% of non-bacterial cases of diarrhea [10]. Reports have collected the percentages of samples in which astrovirus was identified in children with AGE. A study in two Chinese provinces found 1.7% and 4% of positive samples in children aged < 5 years [20] while, in Russia, astrovirus was detected in 2.8% of samples from children aged < 3 years [21]. In France, the percentage of positive samples was 1.8% in children aged < 5 years [5] and an Italian study of children aged < 2 years found a rate of positive samples of 3.1% [22] while another Italian study found a rate of 18.9% in children aged < 3 years [23].

Two Spanish studies in Barcelona in children aged ≥ 5 years with AGE, the first in 1997–2000 and the second in 2016–2017, identified astrovirus in 4.9% and 7.6% of samples analyzed, respectively [3,24].

Several reports on AGE outbreaks of viral etiology in daycare centers with the involvement of HAstVs have been described [15,25,26], but they did not provide specific data on their extent among household contacts.

The AR observed in astrovirus outbreaks vary widely. We found a global AR of 20%, similar to the 17.6% (261/1479) described by Lui et al. [27] in an outbreak in schoolchildren in the Chinese province of Guangxi in 2017 and less than the 70% (7/10) in children aged 7–18 months described by Taylor et al. [26] in an outbreak in a South-African daycare center. Other reports have found smaller AR, ranging between 3.2% and 8.5% [28,29].

The long period between the first case and symptom onset of the other cases (more than one incubation period) in the daycare center might be explained by circulation of the virus among children’s household contacts, as suggested by the two cases at the beginning of the outbreak, or because other members of the daycare center could have been infected asymptomatically.

No increase in the incidence of AGE in the municipality beyond those related to the childcare center was observed. In the first days of the outbreak there was a progressive onset of clinical cases in attenders of the daycare center and their household contacts. Other asymptomatic or symptomatic infections may have been propagated in the community and may have been underreported because healthcare was not required.

A progressive occurrence in cases, as shown by the epidemic curve, is typical of direct person-to-person propagation and suggests that this was the route of transmission [30]. However, fomite involvement cannot be ruled out [31,32].

To avoid the spread of HAstV infections in daycare centers the interruption of person-to-person transmission [33] is important and, therefore, training in infection control procedures including adequate hand hygiene and the cleanliness of surfaces of daycare providers and families of attenders should be promoted [34].

The highest AR was in children aged 1 year, similar to other reports. A study by Olortegui et al. that followed 2082 children from eight countries from 17 days to two years of age found that 26.4% of children had been infected at least once during the first year of life and 35.2% at two years of age [35].

In the outbreak described here there were four asymptomatic astrovirus infections (40% of HAstV detected), which is in agreement with other reports [36].

Norovirus GI was identified in two clinical samples (one in a child aged < 1 year and the other in an adult), which is not surprising given the high frequency of norovirus infection in the community. The prevalence of asymptomatic norovirus infection in children has sometimes surpassed 30% in some industrialized countries [37]. Viral coinfection has been described in isolated cases and outbreaks of AGE [25,38]. In neither of the two norovirus cases in this outbreak was there coinfection.

Our results suggest that the outbreak may be attributed to the circulation of more than one HAstV serotype. The fact that the two cases in which HAstV-8 was identified were asymptomatic and that HAstV-4 was identified in a patient with symptom onset in the initial phase of the outbreak supports the idea that the outbreak was caused by HAstV-4, but given the limited number of samples we cannot state this for sure.

Clinical cases occurred in 26.9% of the homes of both children and workers from the daycare center (16 household contacts of staff or children who become ill and 2 in non-affected people). Propagation of a childcare-associated outbreak in the community has been described for norovirus outbreaks [39,40], but not for astroviruses.

Our study has some limitations. First, no clinical samples were obtained from household contacts. However, the AR of household contacts of clinical cases were significantly higher than the AR of household contacts of non-affected attenders, suggesting that clinical cases in household contacts were related to the childcare-associated outbreak and not to community circulation of these astrovirus strains or other viruses. A second limitation is the long period between the first clinical case, which was not notified, and the start of the investigation, which probably made it difficult to identify other clinical cases. Thirdly, the mildness of the disease might have meant that some persons affected did not seek healthcare, which would have underestimated the overall impact of the illness in the community.

## 5. Conclusions

In conclusion, this outbreak of AGE due to astrovirus in a daycare center caused asymptomatic infections and person-to-person transmission that was not limited to the daycare center where the first clinical cases were notified but extended to household contacts of attenders. Education on hand hygiene and the cleanness of surfaces of daycare providers and family members of attenders should be reinforced when AGE outbreaks occur.

## Figures and Tables

**Figure 1 viruses-13-01100-f001:**
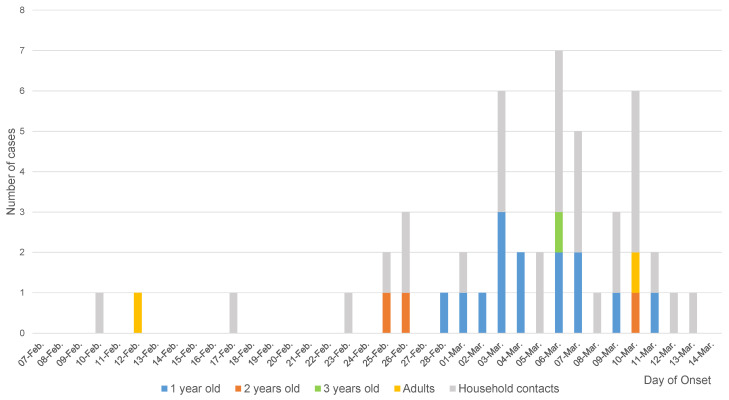
Epidemic curve of cases according to the day of symptom onset.

**Table 1 viruses-13-01100-t001:** Attack rates and risk ratio in day care center attenders and household contacts according to age.

	Age	Affected	Total	AR	RR (95% CI)
Daycare attenders	0 year	0	5	0%	NC
	1 year	14	29	48.3%	1
	2 years	3	28	10.7%	0.22 (0.07–0.69)
	3 years	1	9	11.1%	0.23 (0.03–1.52)
	Adults	2	5	40%	0.56 (0.26–1.23)
Household contacts	0–5	6	22	27.3%	0.82 (0.30–2.20)
	6–10 years	5	15	33.3%	1
	11–25 years	0	11	0%	NC
	26–35 years	5	42	11.9%	0.36 (0.12–1.06)
	36–45 years	13	72	18.1%	0.54 (0.23–1.29)
	46–55 years	0	5	0%	NC
	Total contacts *	29	169	17.2%	

* In two household contacts age was not recorded; NC: Not calculable.

**Table 2 viruses-13-01100-t002:** Samples from exposed people with viruses identified by RT-PCR *.

Sample ID	Clinical Case (Yes/No)	Age	Virus	Date of Symptom onset
1	No	14 months	NoV GI	–
2	No	14 months	HAstV-8	–
3	No	9 months	HAstV-4	–
4	Yes	17 months	HAstV-4	28/2/17
5	Yes	16 months	HAstV-4	6/3/17
6	Yes	14 months	Astrovirus non-typeable	6/3/17
7	Yes	22 months	Astrovirus non-typeable	7/3/17
8	No	21 months	HAstV-4	–
9	Yes	21 months	HAstV-4	4/3/17
10	Yes	36 months	Astrovirus non-typeable	6/3/17
11	Yes	32 years	NoV GI	10/3/17
12	No	34 years	HAstV-8	–

* 7 samples were negative for all viruses screened.

**Table 3 viruses-13-01100-t003:** Attack rates in household contacts according to the clinical status of daycare attenders.

Status of Attenders	Household Contacts			
	Affected	Exposed	AR	RR (95% CI)
Affected	26	35 *	74.3%	30.95 (9.95–96.27)
Non-affected	3	125	2.4%	1

* No information was obtained about involvement in 9 household contacts of attenders affected in the daycare center.

**Table 4 viruses-13-01100-t004:** Attack rates and risk ratios of household contacts by the age of child attenders.

Age of Daycare Center Attenders	Household Contacts			
	Affected	Total	AR	RR (95% CI) *
<1 year	0	16	0%	0.15 (0.01–2.40)
1 year	19	65	29.2%	3.04 (1.51–6.12)
2 years	5	58	8.6%	0.40 (0.16–0.99)
3 years	2	21	9.5%	0.52 (0.13–2.04)
Workers	3	9	33.3%	2.05 (0.76–5.51)

* Each category is compared with the other categories.

## Data Availability

The datasets generated during the current study are available in the Mendeley Data repository (http://dx.doi.org/10.17632/pftbhzznd5.1).

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
