# Peer review of "Human Astrovirus Outbreak in a Daycare Center and Propagation among Household Contacts"

_viruses, 2021, doi:10.3390/v13061100_

Round 1
Reviewer 1 Report
The authors tried to analyze an AGE outbreak due to astrovirus (AstV) in a daycare center and propagation among household contacts. As a result, some AstV genotypes and presumably non-typable AstV were detected in the AGE patients. Moreover, they argued the transmission routes of the viruses. I felt that this study partly contribute to elucidate epidemiology of AstV in elderly homes. However, major concerns were found.
- The authors did not describe typing methods and nucleotide identities of the viruses in Materials/Methods and Results sections. Moreover, they did not describe domestic and other countries' epidemiology of the viruses in Discussion.
- No phylogenetic analysis was performed.
- No description of demographic data including onset rate etc. those were compared with other previous reports.
- Should provide a subheading in the first paragraph of the MM section.
Author Response
Dear reviewer,
We have marked up any revisions to the manuscript using the "Track Changes" function but, in order to that any changes can be easily viewed by the editors and reviewers, we have used red letter-type to highlight changes in the manuscript
- The authors did not describe typing methods and nucleotide identities of the viruses in Materials/Methods and Results sections. Moreover, they did not describe domestic and other countries' epidemiology of the viruses in Discussion.
RESPONSE: We now state “NucliSENS® easyMAG® (Biomerieux) was used to obtain genetic material. RT-PCR was used to identify the main viruses causing AGE (norovirus GI and GII, rotavirus, adenovirus, astrovirus and sapovirus) using the Allplex GI-Virus Assay (Seegene Inc.). Astrovirus genotyping was performed by sequencing the RT-PCR product according to a previously described protocol [19], and compared with Genbank sequences using Blastn analysis, which was also used to compare isolates within the same genotype. “
And
“Sequence comparison with HAstV reference strains showed a 98.52% and 98.52% nucleotide similarity with EF138831.1 and MG970100.1, respectively. All sequences belonging to the same serotype were identical. Sequences showed the closest similarity to published sequences.” (Line 150).
We also comment in the Discussion on the epidemiology of human astroviruses in Spain and other countries and the incidence of acute gastroenteritis due to astrovirus in the community internationally and in Spain.
- No phylogenetic analysis was performed.
RESPONSE: Sequencing was performed as a mean of typing the astrovirus positive specimens using the common well-established typing method first described by Noel et al (1995), which has been extensively used in the astrovirus field to type classic human astroviruses since its publication and also in recent studies in different parts of the world (see references). Phylogenetic analysis was not the aim of the study but, blastn analysis showed the closest similarity with sequences isolated. This information has been added to the revised version of the manuscript.
References:
Chhabra P, Payne DC, Szilagyi PG, Edwards KM, Staat MA, Shirley SH, Wikswo M, Nix WA, Lu X, Parashar UD, Vinjé J. Etiology of viral gastroenteritis in children <5 years of age in the United States, 2008-2009. J Infect Dis. 2013 Sep 1;208(5):790-800. doi: 10.1093/infdis/jit254.
Wu L, Teng Z, Lin Q, Liu J, Wu H, Kuang X, Cui X, Wang W, Cui X, Yuan ZA, Zhang X, Xie Y. Epidemiology and Genetic Characterization of Classical Human Astrovirus Infection in Shanghai, 2015-2016. Front Microbiol. 2020 Sep 25;11:570541. doi: 10.3389/fmicb.2020.570541.
Biscaro V, Piccinelli G, Gargiulo F, Ianiro G, Caruso A, Caccuri F, De Francesco MA. Detection and molecular characterization of enteric viruses in children with acute gastroenteritis in Northern Italy. Infect Genet Evol. 2018 Jun;60:35-41. doi: 10.1016/j.meegid.2018.02.011.
van der Doef HP, Bathoorn E, van der Linden MP, Wolfs TF, Minderhoud AL, Bierings MB, Wensing AM, Lindemans CA. Astrovirus outbreak at a pediatric hematology and hematopoietic stem cell transplant unit despite strict hygiene rules. Bone Marrow Transplant. 2016 May;51(5):747-50. doi: 10.1038/bmt.2015.337.
- No description of demographic data including onset rate etc. those were compared with other previous reports.
RESPONSE: Data on attack rates from previous studies of astrovirus outbreaks have been added in the Discussion section of the revised manuscript and compared with those obtained in our study.
- Should provide a subheading in the first paragraph of the MM section.
RESPONSE: We have added a subtitle to the first paragraph of the Material and Methods section.

Reviewer 2 Report
The authors describe an outbreak of acute gastroenteritis (AGE) in a daycare in Catalonia, Spain. The outbreak period lasted over a month (Feb 1 – Mar 12 2017) with 123 children and workers exposed. Epidemiological data including demographics, the date of onset, end and type of clinical manifestations, the number of people living in the home of each case, as well as whether they became ill, was gathered using a self-administered questionnaire filled out by the parents of the children or the daycare workers. Overall 245 subjects completed questionnaires. There was no significant difference in incidence between males and females; however, 2-year-old children had a statistically significant lower AR than other age groups. Stool samples collected from both symptomatic and asymptomatic, exposed individuals were RT-PCR for NoV GI and HAstV. Additionally, the authors report a strong statistical association between clinical manifestation of attenders and household contacts’ ARs and statistically significant difference in the AR of household contacts of 1- and 2-year-old children. While outbreaks of viral AGE in daycare centers have been previously reported, this study provides the field with a descriptive report where the incidence of household contacts are also reported.
Major Comment:
- The information regarding the stool samples appears to be incomplete. The methods section indicates19 samples taken (13 affected, 6 non-affected) only 12 samples listed in table 2. What were the results of the other 7 samples? Were any longitudinal samples taken? Additionally, the methods sections states stool samples were collected between March 13-15, this would be near the end of the outbreak. Were any clinical samples taken during the outbreak period? This might be able to narrow which HAstV serotype was responsible for the outbreak given that more than one serotype was identified.
Minor Comment:
- The manuscript would benefit from editorial revisions for grammar. Also, there is inconsistency in the abbreviation of the HAstV serotypes in the text from the introduction to the results and inn Table 2.
Author Response
Dear reviewer,
We have marked up any revisions to the manuscript using the "Track Changes" function but, in order to that any changes can be easily viewed by the editors and reviewers, we have used red letter-type to highlight changes in the manuscript.
- The information regarding the stool samples appears to be incomplete. The methods section indicates19 samples taken (13 affected, 6 non-affected) only 12 samples listed in table 2. What were the results of the other 7 samples? Were any longitudinal samples taken? Additionally, the methods sections states stool samples were collected between March 13-15, this would be near the end of the outbreak. Were any clinical samples taken during the outbreak period? This might be able to narrow which HAstV serotype was responsible for the outbreak given that more than one serotype was identified.
RESPONSE: Viruses were identified in only 12 of the 19 samples analyzed. The remaining 7 samples were negative for all viruses screened for. A footnote has been added to Table 2 for the sake of clarity.
No longitudinal samples were collected from the subjects studied. This has been specified in the Study design in the revised version.
Since the date of notification of the outbreak to the epidemiological surveillance unit was late (25 days after the first case), samples could only be collected during the late phase of the outbreak.
We now state in the Discussion " The fact that the two cases in which HAstV-8 was identified were asymptomatic and that HAstV-4 was identified in a patient with symptom onset in the initial phase of the outbreak supports the idea that the outbreak was caused by HAstV-4, but given the limited number of samples we cannot state this for sure.”
- The manuscript would benefit from editorial revisions for grammar. Also, there is inconsistency in the abbreviation of the HAstV serotypes in the text from the introduction to the results and inn Table 2.
RESPONSE: The manuscript has been revised by a native English speaker. We have also corrected the errors in the abbreviation of the HAstV serotypes.

Round 2
Reviewer 1 Report
The authors well addressed for the comments by the Editor.
One minor issue.
The authors should use "nucleotide identity" instead of "nucleotide similarity".
Author Response
- The authors should use "nucleotide identity" instead of "nucleotide similarity".
RESPONSE: Thank you for your suggestion. We have made the change proposed.